# Molecular identification of organic vapors driving atmospheric nanoparticle growth

Claudia Mohr[1]*, Joel A. Thornton[2]*, Arto Heitto[3], Felipe D. Lopez-Hilfiker[2,7], Anna Lutz[4], Ilona Riipinen[1], Juan Hong[5], Neil M. Donahue [6], Mattias Hallquist [4], Tuukka Petäjä [5], Markku Kulmala[5] & Taina Yli-Juuti[3]*

Particles formed in the atmosphere via nucleation provide about half the number of atmospheric cloud condensation nuclei, but in many locations, this process is limited by the growth of the newly formed particles. That growth is often via condensation of organic vapors. Identification of these vapors and their sources is thus fundamental for simulating changes to aerosol-cloud interactions, which are one of the most uncertain aspects of anthropogenic climate forcing. Here we present direct molecular-level observations of a distribution of organic vapors in a forested environment that can explain simultaneously observed atmospheric nanoparticle growth from 3 to 50 nm. Furthermore, the volatility distribution of these vapors is sufficient to explain nanoparticle growth without invoking particle-phase processes. The agreement between observed mass growth, and the growth predicted from the observed mass of condensing vapors in a forested environment thus represents an important step forward in the characterization of atmospheric particle growth.

---

[1] Department of Environmental Science and Analytical Chemistry, Stockholm University, 11418 Stockholm, Sweden. [2] Department of Atmospheric Sciences, University of Washington, 98195 Seattle, WA, USA. [3] Department of Applied Physics, University of Eastern Finland, 70211 Kuopio, Finland. [4] Department of Chemistry and Molecular Biology, University of Gothenburg, 41296 Gothenburg, Sweden. [5] Institute for Atmospheric and Earth System Research/Physics, Faculty of Science, University of Helsinki, 00014 Helsinki, Finland. [6] Center for Atmospheric Particle Studies, Carnegie Mellon University, 15213 Pittsburgh, PA, USA. [7] Present address: Tofwerk AG, 3600 Thun, Switzerland. *email: claudia.mohr@aces.su.se; thornton@atmos.uw.edu; taina.yli-juuti@uef.fi

For atmospheric particles to serve as seeds for cloud condensation, they typically need to be larger than a few tens of nanometer in size[1]. Consequently, in many locations, growth is the limiting factor for particles formed in the atmosphere via nucleation to become active as cloud condensation nuclei (CCN). It is estimated that newly formed particles provide about half of the number of atmospheric CCN[2,3]. Condensing organic vapors represent an important, if not dominant pathway for particle growth[4–8]. Identification of these vapors and their sources is fundamental to developing robust parametrizations of CCN formation needed to better understand aerosol–cloud interactions, which are one of the most uncertain aspects of anthropogenic climate forcing[9,10].

The terrestrial biosphere is an important source of condensable organic vapors capable of driving growth[5,11–13]. Extremely low volatility organic compounds (ELVOC) play a crucial role in early (<5 nm) particle production in the field and in the laboratory[14–17]. Many ELVOC are highly oxygenated molecules (HOM) formed via autoxidation of biogenic vapors[18,19]. However, recent laboratory data suggest that organic compounds other than the ELVOC are needed to explain the majority of growth from 5 to 50 nm[15,20]. Moreover, the composition and properties of such vapors have remained unresolved[5], along with the associated mechanisms of gas-to-particle conversion in the atmosphere, in part because these particle sizes remain outside the measurable range of field deployable aerosol mass spectrometers[21].

Here, we identify the molecular formulae of a large distribution of organic vapors measured in a forested environment. Importantly, we quantitatively show that this distribution together with the properties inferred from the determined molecular compositions can explain atmospheric nanoparticle growth from 3 to 30 nm at this location via a first-principles model of condensational growth. Modeled particle growth rates are entirely independent of measured particle growth rates, as they are based on the measured organic vapor concentrations and their inferred volatilities. Our agreement in measured nanoparticle mass growth, and the mass of available condensing vapors, based on direct ambient observations, therefore represents a key step forward in our understanding of particle growth to sizes relevant for CCN formation.

## Results

### Molecular identity and volatility of organic vapors observed over the boreal forest.

We obtained the molecular identity and real-time, simultaneous observations of more than 1000 oxygenated volatile organic compounds (OVOC) over the boreal forest of Finland utilizing the University of Washington's time-of-flight chemical ionization mass spectrometer with iodide-adduct ionization[22,23] (I-TOF-CIMS, Methods) deployed in Hyytiälä, Finland, during Spring 2014. Of these, 618 ions had molecular compositions comprising 1–30 carbon atoms and up to 17 oxygen atoms, and an additional 473 ions also contained 1–2 nitrogen atoms (representative mass spectrum in Supplementary Fig. 1). Following previous work[17,24], we estimate the saturation concentration ($C_{sat}$) for each unique molecular composition based on the number of carbon, oxygen, and nitrogen atoms in that molecule as proxies for the distribution of functional groups. The $C_{sat}$ is a measure of a compound's volatility, which together with gas-phase concentrations is the driving force behind particle growth via condensation, and yet the distribution of vapor mass in a $C_{sat}$ space relevant to nanoparticle growth has generally been lacking. Our quantitative, speciated measurements of organic vapor composition from the real atmosphere thus provide a key test of molecular-level mechanistic descriptions of nanoparticle growth in a forest environment.

We show the distribution of observed gaseous OVOC and their concentrations measured by the I-TOF-CIMS in Fig. 1a versus both the number of oxygen and carbon atoms per molecule, along with the inferred volatility distribution. To obtain vapor concentrations from mass spectral ion signals, we assume a maximum ionization efficiency based on an experimentally verified collision-limited rate of ion adduct formation[25] (see the Methods section). This assumption produces a conservative (lower-limit) estimate of mass concentrations and thus of mass fluxes to particles. Compounds with ten or fewer carbon atoms, and eight or fewer oxygen atoms have the highest mass concentrations. These compounds mostly fall into volatility bins with $C_{sat} \geq 1\ \mu g\ m^{-3}$ ($N = 2 \times 10^9$ molecules $cm^{-3}$) and are thus semi-volatile organic compounds (SVOC). However, we also observe substantial signals from LVOC or ELVOC, respectively. Less functionalized and thus more volatile OVOC are not detected by I-TOF-CIMS[23].

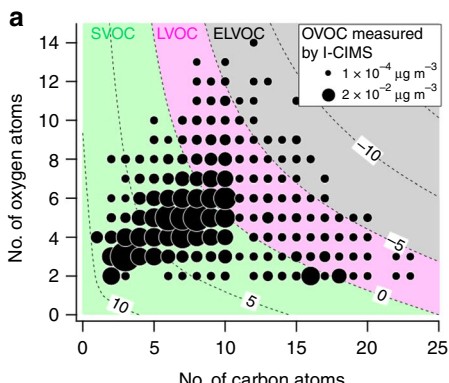
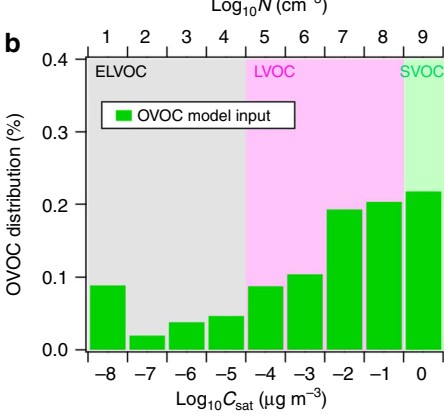

**Fig. 1** Distribution of measured oxygenated volatile organic compounds (OVOC). **a** OVOC (including N-containing compounds) measured by a time-of-flight chemical ionization mass spectrometer with iodide-adduct ionization (I-TOF-CIMS) are plotted as a function of the number of carbon and oxygen atoms per molecule. The size of the circles is proportional to the total measured mass concentrations for that composition (average over 13 new particle formation events). Contour lines indicate the log of the saturation concentration ($C_{sat}$) as a function of oxygen and carbon number. **b** The same OVOC compounds as shown in Fig. 1a binned into a volatility basis set (VBS) as input into the Model for Acid–Base Chemistry in Nanoparticle Growth (MABNAG). The model input consisted of 15 bins ($\log_{10}C_{sat} = -14$ to 0). The figure bin with $\log_{10}C_{sat} = -8$ is the sum of MABNAG bins $\log_{10}C_{sat}$ −14 to −8

We recorded this distribution of OVOC composition and concentrations in situ on an hourly time basis across all new particle formation (NPF) events during our measurement campaign when the I-TOF-CIMS was operational (13 in total), thereby providing an opportunity to test whether this observed distribution is sufficient to explain the condensational growth rates of newly formed atmospheric particles from 3 nm up to 50 nm. We binned the gaseous OVOC concentrations measured by I-TOF-CIMS during NPF into a volatility-basis set (VBS) using an updated parametrization based on the number of carbon and oxygen atoms in a given compound[24] (see the Methods section). Importantly, in contrast to previous studies[15,20], the $C_{sat}$ parametrization was neither tuned to our data set nor did we attempt to fill any observational gaps in the distribution by extrapolation. Therefore, the parametrization is completely independent of the compounds measured here, and the distribution of mass in the volatility space is entirely determined by the measured abundance of individual compositions. We used volatility bins with $C_{sat} \le 1\ \mu g\ m^{-3}$ (Fig. 1b) as input to a model of nanoparticle growth (Model for Acid–Base Chemistry in Nanoparticle Growth, MABNAG[26] (Methods)) to simulate particle growth rates observed during NPF events. The more volatile SVOC ($C_{sat} > 1\ \mu g\ m^{-3}$) never reached a gas-phase concentration (saturation ratio) high enough to contribute significantly to nonreactive condensation. We assumed the condensing and evaporating OVOC to have an accommodation coefficient of unity.

**Agreement between predicted and observed particle growth rate**. In Fig. 2, we show the predicted particle growth for the NPF event from April 16, 2014, and the evolution of particle size distributions measured independently by a differential mobility particle sizer (DMPS). Considering the order of magnitude certainty of $C_{sat}$ parametrizations[27], there is very good agreement between the modeled and the centroid of measured growth rates, indicating that the distribution of compounds measured by I-TOF-CIMS and their inferred volatilities fully explain particle growth rates at that location. The I-TOF-CIMS-constrained predictions of particle growth and measured growth versus time are shown for all events sampled during the April–May 2014 intensive campaign in Supplementary Fig. 2. Ranges given in Fig. 2 encompass uncertainties related to $C_{sat}$ estimates, measured concentrations of OVOC used in the model, and start times of the NPF event (Methods).

As shown in Fig. 3, as well as in Supplementary Fig. 2, modeled growth rates (GR), using the input from I-TOF-CIMS, and measured growth rates (averages over entire NPF events for both) are in remarkable agreement across several weeks and multiple NPF events, given the natural variability in ambient data and the fact that no artificial tuning was applied to the $C_{sat}$ derived from observed compositions for use in the model calculations: For the majority of NPF events, the ratio of modeled and measured growth rates lies between 0.7 and 2.1 (median ratio all events: 2.1). The data points in Fig. 3 depicting overestimations of a factor of 3–6 correspond to the growth events between May 7 and May 18. The event on May 7 exhibits a rather unclear start time, and the event on May 14 represents a short or only partial observation of a growth event[28]. The May 17 and 18 events took place while there was a clear accumulation mode present, due to advection of polluted air masses. Uncertainties stemming from $C_{sat}$ parametrizations significantly affect the mean bias between modeled and measured growth rates as illustrated by Supplementary Figs. 3 and 4. Indeed, the degree of agreement we achieve with ambient measurements suggests promise for testing the applicability of $C_{sat}$

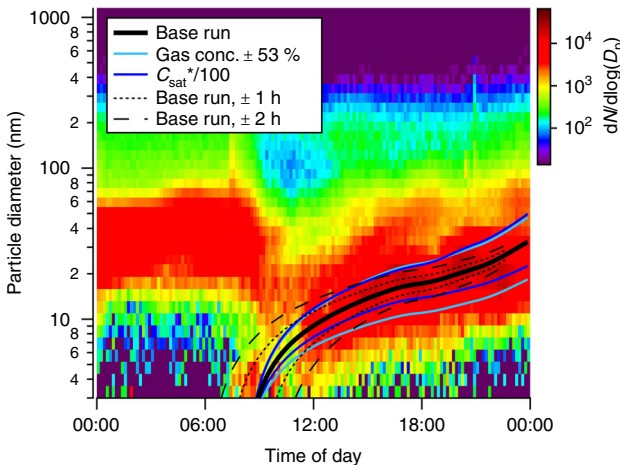

**Fig. 2** Modeled and measured particle growth. Modeled growth rate for the new particle formation event on April 16, 2014, using oxygenated volatile organic compounds (OVOC) measured by the time-of-flight chemical ionization mass spectrometer with iodide-adduct ionization (I-TOF-CIMS) as model input, on top of size distributions measured by a differential mobility particle sizer (DMPS)

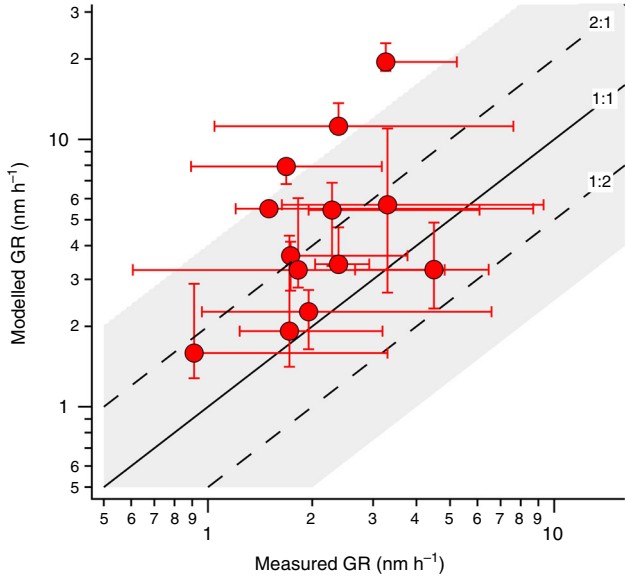

**Fig. 3** Modeled vs measured growth rates (GR) for all 13 new particle formation events (3–30 nm). Modeled GR are calculated over the growth from 3 to 30 nm by a linear fit to the diameter as a function of time. The error bars for the modeled GR represent the variation in GR if it is calculated as a fit to different size ranges (3–5 nm, 5–10 nm, 10–20 nm, and 20–30 nm). The measured GR are calculated from particle size distributions by a linear fit to the nucleation mode peak diameter as a function of time. The error bars for the measured GR represent the variation in the growth rate when the linear fit is performed on mode peak diameters from different time intervals of the observed nanoparticle growth

parametrizations derived from laboratory experiments or group-contribution methods. We have no indication of a systematic positive bias of the measured OVOC as model input (Methods), or proof of an accommodation coefficient lower than unity[29], but since these parameters also affect modeled growth rates we have added results from these model runs to Supplementary Figs. 3 and 4.

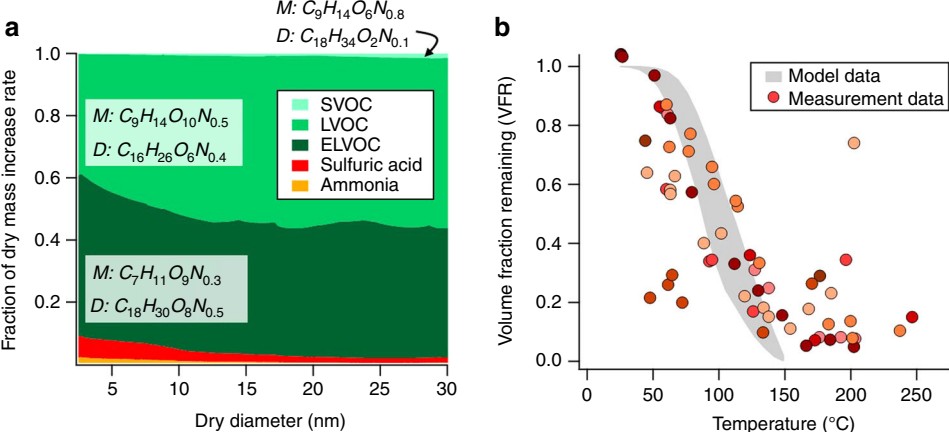

**Fig. 4** Particle-phase composition and volatility. **a** Modeled average contribution (all new particle formation (NPF) events) of 17 volatility bins lumped to semi-volatile organic compounds (SVOC), low volatility organic compounds (LVOC), and extremely low volatility organic compounds (ELVOC) to dry particle mass increase as a function of particle size with mass-weighted mean molecular formulae for compounds with <11 carbon atoms ("monomers", M) and >10 carbon atoms ("dimers", D). **b** Volume fraction remaining (VFR) of particles with diameter 30 nm measured by a thermodenuder (TD) for all NPF events (circles). The different colors represent different NPF days. Shaded area: modeled VFR curve using the particle volatility distribution as given by the output of the Model for Acid–Base Chemistry in Nanoparticle Growth (MABNAG), all NPF events

Our observed gas-phase OVOC distribution together with reasonable and internally consistent estimates of $C_{sat}$ can fully reproduce the time evolution of nanoparticle growth occurring over the 5–10-h periods typical of multiple NPF events (see Supplementary Fig. 2 for all events). This near closure with a condensation-only model therefore provides constraints on the importance of other processes in driving the growth up to 50 nm in a pine forest setting. For example, in the growth-rate calculations, we assume that once in the particle phase, the compounds do not further react in a way that significantly changes the mass uptake. Moreover, our analysis indicates only a minor importance of particulate-phase organic salt formation, a mechanism specifically included in MABNAG[26]. The model does not include other possible particulate-phase accretion reactions[30] to form high-molecular-weight compounds[31–33]. Such accretion reactions almost certainly occur and may be important for sustaining mass over the multiday lifetime of a submicron particle, but they are not required to explain the initial growth to CCN sizes of newly formed particles.

**Predicted particle-phase composition and volatility distribution.** The model-predicted particle-phase composition (Fig. 4a) is largely independent of uncertainties in modeled growth rate stemming from measured gas-phase concentrations, $C_{sat}$ parametrization, or mass accommodation coefficient (Supplementary Fig. 3). Such a robust result implies that LVOC condensation dominates nanoparticle growth, even at the relatively early stages (3–5 nm). Our observations thus show for ambient air that LVOC contributions are indeed needed to explain observed particle growth rates, which until now had only been invoked in laboratory studies[15,20]. The I-TOF-CIMS measurements constrain the available condensable vapors and suggest that the dominant components of growing particles in the 3–30 nm range are monomers of monoterpene-oxidation products, with a mass-weighted mean of 7–9 carbon atoms and 9–10 oxygen atoms. We observe a secondary contribution from likely gas-phase accretion products, with a mass-weighted mean of 16–18 carbon atoms and 6–8 oxygen atoms. Many of these compounds are similar to those observed or described previously in field data analyses and laboratory chamber studies of monoterpene oxidation[14,15,17,34,35]. Moreover there is a non-negligible but lesser contribution from nitrogen-containing compounds in daytime particle growth to

50 nm at this location with 0.3–0.5 mass-weighted nitrogen atoms in the ELVOC and LVOC bins. Taken as a whole, these insights corroborate the dominant role of emissions from the biosphere for particle growth to CCN sizes under relatively unpolluted ($NO_x$ < 1 ppbv) conditions.

Only a fraction of both ELVOC and LVOC compounds have specific molecular formulae corresponding to HOM monomers previously reported[15], which is not surprising given the range of compounds the I-TOF-CIMS is able to measure compared with the nitrate CIMS used for HOM measurements[36]. Autoxidation processes that lead to HOM are likely relevant for many of the compounds we do observe. We observe significant contributions from HOM dimers, as well as non-HOM dimers we reported earlier[17]; however, the majority of mass is contributed to the growing particles by monomers. We therefore conclude that it is the monomers (which may or may not be formed via autoxidation) of sufficiently low volatility that drive a majority of particle growth in this setting, and that dimer formation in the gas or particle phase plays a lesser role in the growth to 50 nm. In our model, the SVOC and LVOC are calculated to make smaller contributions at the very early stages of particle growth due to the influence of the Kelvin effect, hence the relative contribution of the ELVOC is largest at the smallest particle sizes. Other work suggests that the ELVOC monomers and dimers play a dominant role in particle nucleation itself[37]. ELVOC also remain significant contributors to particle mass (30%) up to sizes of 50 nm.

**Closure of predicted and measured particle-phase volatility.** As an independent check on the volatility distribution driving the predicted particle growth, we compare the predictions of particle volatility at a size of 30 nm, computed in MABNAG from the observationally constrained VBS, to in situ observations of temperature-dependent evaporation, which provide a measure of effective volatility. The observed volume fractions remaining (VFR) of 30 nm particles as a function of temperature, between ambient and 250 °C, as measured by a thermal denuder (TD, Methods) during 11 out of 13 NPF events of the study period are shown in Fig. 4b. Also shown in Fig. 4b are the corresponding TD model[38] predicted VFR curves arising from the volatility distribution of particulate compounds calculated with MABNAG to be present in 30 nm particles for the same day and NPF event (Methods). We obtain broad agreement of measured and

modeled particle volatility without needing to invoke additional volatility-lowering processes, such as particle-phase reactions. Moreover, a similar comparison of predicted particle volatility to that derived from a separate set of measurements by an I-TOF-CIMS with a Filter Inlet for Gases and AEROsols (FIGAERO) at the same location and season (Methods), but a different year (2013), also shows very good agreement (Supplementary Fig. 5). Such closure of model output based on ambient gas-phase observations in 1 year, with particle volatility measurements in multiple years, underlines the robustness of results presented here. We note that there is little evidence to suggest large variation in the seasonal drivers of NPF and growth events in this remote boreal forest setting[39].

## Discussion

By identifying the molecular composition and directly constraining the absolute and relative abundances of hundreds of SVOC, LVOC, and ELVOC, we quantitatively explain atmospheric nanoparticle growth in a remote boreal forest environment during spring time. This allows us to draw conclusions regarding the precursors and processes that drive nanoparticle growth from 5 to 30 nm in diameter in similar environments. We captured the observed particle growth rate in multiple NPF events based on the quantitative measurements of OVOC concentrations with our growth-model construct that neglects particle-phase chemistry of organic compounds. Our analysis confirms that LVOC are responsible for a majority of newly formed particle growth, as suggested previously from controlled laboratory studies of particle growth[15,20] with an important contribution from ELVOC, and a much smaller, but not negligible contribution from SVOC. Moreover, the measurements of molecular composition strongly point to a dominant role for monoterpene oxidation products in driving growth, with minimal evidence for isoprene or sesquiterpene contributions. While particle-phase chemistry and viscosity may control particle volatility on longer timescales (days), the early growth period we simulate here suggests that accretion or salt formation chemistry, to the extent it occurs, is either initially reversible or unimportant to particle growth over the first 5–10 h of new particles' existence.

With the availability of highly chemically and temporally resolved data sets such as ours likely to increase in the near future, our study will help constrain and understand growth also in other environments than the one described here. Explaining the growth of newly formed particles to CCN relevant sizes is more critical than determining nucleation rates of new particles for accurately modeling the distribution of CCN[7]. The same growth mechanism should also apply to primary nanoparticles, such as combustion emissions. The importance of LVOC and even SVOC condensation to nanoparticle growth also indicates that anthropogenic VOC oxidation products, which tend to be more commonly in the LVOC and SVOC categories, could also play a role in particle growth to CCN relevant sizes in highly polluted regions where those precursors are significant. The insights provided herein allow for the development of robust parametrizations of new particle growth to CCN relevant sizes driven by monoterpene oxidation products. Such parametrizations are a necessity for improved assessments of CCN perturbations and associated uncertainty in aerosol climate forcing.

## Methods

**Chemical ionization mass spectrometer (I-TOF-CIMS).** The data for the present study were acquired during April 11–June 3, 2014 at the Station for Measuring Ecosystem-Atmosphere Relations II (SMEAR II) site of the University of Helsinki situated in the Hyytiälä Forestry Field Station in Hyytiälä, Finland (61°50′51″N; 24°17′42″E)[40] with a time-of-flight chemical ionization mass spectrometer utilizing iodide-adduct ionization (I-TOF-CIMS)[17,41]. The instrument consists of a reduced-pressure ion-molecule-reaction (IMR) chamber coupled to an

atmospheric pressure interface high-resolution time-of-flight mass spectrometer (Tofwerk AG, Thun, Switzerland). Measurements were performed on the top platform of a 35 m walkup tower, where the instrument was housed in a temperature-controlled weather-proof container. Iodide-adduct ionization was used to measure oxygenated organic compounds with a mass accuracy of ≤20 ppm. Iodide ions were generated via a permeation tube filled with methyl iodide, over which a 2 slpm flow of ultrahigh purity $N_2$ was passed and then guided through a Po-210 ion source into the IMR[23]. Gases were drawn at 22 standard liters per min through a 19-mm outer diameter PTFE inlet extending 1 m horizontally off the tower. Gas-phase composition was determined at 10 Hz for 45-min periods. Gas-phase backgrounds were determined by overflowing the critical orifice at the inlet to the IMR with UHP $N_2$ for 10 s every 5 min during the gas-phase measurement period.

**Conversion of ion signal to mass concentration.** For the conversion of I-TOF-CIMS signal (Hz) to atmospheric mass concentrations, we used the collision-limit value of 22 counts $s^{-1}$ $ppt^{-1}$ per MHz of reagent ion determined for iodide-adduct formation in this same instrument[25]. In earlier publications[23,42], we show the relationship between sensitivity and molecular mass of compounds based on calibrations of various compounds (with sensitivity being a combination of ionization efficiency and transmission of compounds through the mass spectrometer): for compounds with mass-to-charge (m/z) ratios larger than 200, their sensitivity approaches the value of the empirically determined collisional limit used in our analysis.

**MABNAG and VBS parametrization.** Nanoparticle growth was simulated using the particle growth model MABNAG (Model for Acid–Base Chemistry in Nanoparticle Growth[26]). MABNAG is a monodisperse growth model that combines a particle's internal acid–base chemistry and condensational growth. The growth of particle size and changes in particle composition were calculated based on measured ambient temperature, relative humidity (RH), and gas-phase concentrations of ammonia, sulfuric acid, and oxygenated organic compounds. The organics detected in the gas phase with the I-TOF-CIMS were grouped into a 15-bin volatility basis set (VBS) based on their saturation concentrations ($C_{sat}$), and each bin was presented as an organic model compound. The organic model compounds had $C_{sat}$ between $10^{-14}$ and $10^0$ µg $m^{-3}$ (defined at 300 K) with tenfold differences. All organic compounds with $C_{sat}$ lower than $3 \times 10^{-14}$ µg $m^{-3}$ were included in the least volatile model compound, and all organic compounds with $C_{sat}$ higher than $3 \times 10^0$ µg $m^{-3}$ were neglected in the growth simulations. All organics were assumed to be non-reacting in the particle phase (no dissociation). The molecular mass and molecular volume of each organic model compound were calculated as gas-phase concentration-weighted averages over the properties of the compounds grouped in the model compound.

The $C_{sat}$ values for individual organic compounds were calculated based on the number of oxygen, carbon, and nitrogen atoms in the compound using the following parametrization:

$$\log(C_{sat}) = (n_0 - n_C)b_C - (n_O - 3n_N)b_O - 2\frac{(n_O - 3n_N)n_C}{(n_C + n_O - 3n_N)}b_{CO} - n_N b_N \quad (1)$$

where $n_0 = 25$, $b_C = 0.475$, $b_O = 0.2$, $b_{CO} = 0.9$, and $b_N = 2.5$. $n_C$, $n_O$, and $n_N$ are the number of carbon, oxygen, and nitrogen atoms in the compound, respectively. This is an updated version of the parametrization by Donahue et al.[24], modified based on the saturation concentrations of HOM detected by Tröstl et al.[15]. The modification reflects the extensive presence of −OOH functional groups in HOM formed via autoxidation, which have a much smaller effect on volatility than the −OH and =O functional groups presumed to dominate SOA in the earlier 2D-VBS formulation. For comparison, the growth simulations were also performed using $C_{sat}$ values calculated based on the original parametrization by Donahue et al.[24], and a modified version (different from ours) of the original parametrization used in a very recent publication[20]. The growth simulated based on these three versions of $C_{sat}$ parametrization are shown in Supplementary Fig. 4. The parametrization gives $C_{sat}$ values at a temperature of 300 K. In our study, the organics were first grouped based on their $C_{sat}$ calculated at 300 K, and the temperature dependence of $C_{sat}$ was then accounted for by calculating the $C_{sat}$ value of each model compound at ambient temperature based on the method by Epstein et al.[43]:

$$C_{sat}(T) = C_{sat}(300 \text{ K}) \exp\left(\frac{\Delta H^{VAP}}{R}\left(\frac{1}{300\text{K}} - \frac{1}{T}\right)\right) \quad (2)$$

where $T$ is the temperature in Kelvin, $C_{sat}$ (300 K) is the saturation concentration at 300 K, $R$ is the gas constant, and $\Delta H^{VAP}$ is the vaporization enthalpy. A constant density of 1500 kg $m^{-3}$ and a surface tension of 0.03 N $m^{-1}$ were assumed for the particles.

$$\Delta H^{VAP} = -11 \log_{10} C_{sat}(300 \text{ K}) + 129 \quad (3)$$

Particle growth was simulated for each new particle formation (NPF) event individually observed during the campaign. For each simulation time step, the input values for the model ($T$, RH, and gas concentrations) were interpolated from the two measurement points closest in time. Concentration measurements for ammonia and sulfuric acid were missing for some NPF events. In these cases, daytime averages (8 am–6 pm) over the whole measurement period were used. The

initial composition of the particle in the model simulations was 40 sulfuric acid molecules and a corresponding amount of water and ammonia according to their gas-particle equilibrium constrained by ambient gas-phase observations. With this assumption, the diameter of the particle at the start of the simulation was ~2 nm. The starting times for the simulations were chosen as the times during an NPF event when the total concentration of 3–5 nm particles measured by the DMPS was highest. Each simulation stopped at 99400 s or earlier in case the particle diameter reached 52 nm.

**Uncertainties of growth rates.** In addition to using different $C_{sat}$ parametrizations (see MABNAG and VBS parametrization), we also investigated the sensitivity of the simulated growth on the uncertainty in measured OVOC concentrations, on $C_{sat}$ (factor 100), on shifting the start time of the simulation by ±1 h, and on using a mass accommodation coefficient ($\alpha_M$) = 0.5 (Fig. 2; Supplementary Figs. 2–4). The uncertainties were estimated separately for both I-TOF-CIMS measurements and $C_{sat}$, and these estimations were used as boundary conditions for the simulations. Using $\alpha_M < 1$ allows to simulate particle-phase diffusion limitations trapping semi-volatile compounds or kinetic limitations to growth, but despite the improvement in growth rate predictions, there is no scientific proof for $\alpha_M$ lower than unity. Uncertainties of OVOC mass concentrations were calculated following the procedure detailed in Thompson et al.[44]: In general, gas-phase concentrations of the I-TOF-CIMS are determined as the difference between the signal and background multiplied by sensitivity. Each of the three components has an accuracy and a precision contributing to the overall uncertainty. For the data presented here, signal accuracy (determined from regular calibrations using formic acid) and signal precision (standard deviation of steady signal) were 17 and 9%, respectively; background accuracy and precision (determined using the background measurements regularly performed during the campaign) 40 and 12%, respectively; sensitivity accuracy and precision (determined from repeated calibrations of formic acid) 18 and 20%, respectively, resulting in an overall uncertainty of 53%.

The modeled growth rate for each NPF in Fig. 3 is the value calculated over the growth from 3 to 30 nm by a linear fit to the diameter as a function of time. This size range was selected to correspond to the size range for which the growth rate could be calculated from the measurements. The error bars represent the variation of the growth rate if the value is calculated as a fit to different size ranges (3–5 nm, 5–10 nm, 10–20 nm, and 20–30 nm) and, therefore, arise from the variations both with particle size and in time. The measured growth rates were calculated from particle-size distributions by a linear fit to the nucleation mode peak diameter as a function of time[45]. Variation in the measured growth rate within one NPF event was estimated as the variation in the growth rate when the linear fit was performed on mode peak diameters from different time intervals of the observed nanoparticle growth.

**Thermodenuder and thermodenuder model.** Particle composition predicted with MABNAG was compared with the volatility of 30 nm particles measured with a Volatility Tandem Differential Mobility Analyzer (VTDMA)[46]. In the VTDMA, a monodisperse particle sample is selected with a differential mobility analyzer, the sample is heated by a thermodenuder, and the remaining particle-size distribution is measured with a second differential mobility analyzer and a condensation particle counter. This gives the volume fraction remaining (VFR) of particles upon heating to specific temperatures. Comparison of the measured particle volatility to the modeled particle composition was done by applying a kinetic evaporation model to the modeled particle composition at 30 nm to simulate particle-size reduction upon heating[38]. In these evaporation simulations, the mass of sulfate and ammonium ions was summed and their evaporation was modeled as ammonium sulfate[46]. The same vaporization enthalpies were used for organics as in the growth model.

**Filter Inlet for Gases and AEROsols (FIGAERO) with impactor.** In 2013, the same TOF-CIMS as used here was deployed at the same forest station in Finland, utilizing acetate as a reagent ion[17]. The instrument was equipped with a Filter Inlet for Gases and AEROsols (FIGAERO)[22]. The FIGAERO adds the possibility of particulate OVOC analysis by means of CIMS: while gas-phase compounds are measured in the TOF-CIMS, particles are collected on a Teflon filter for a period of 20 min. The gas-phase measurement is then stopped, and particles are desorbed from the filter via a gradually heated (from ambient temperature to 200 °C in 20 min) stream of $N_2$. The evaporated compounds are analyzed in the TOF-CIMS. After the heating cycle, the FIGAERO switches back to gas-phase measurement and particle deposition. For the Supplementary Fig. 5, we used data from the NPF event on May 6, 2013, when we had installed a Microorifice Uniform Deposit Impactor (MOUDI)[47] at the top of the FIGAERO particle-phase inlet. The theoretical particle cut-size of the MOUDI using all stages is 56 nm, which allowed us to exclude particles with a diameter >56 nm from the analysis, and to focus on the particles formed during the NPF event. However, flow issues likely moved the cut-size to an (unknown) larger diameter. This is likely the reason for the bigger fraction of SVOC measured in the particle phase (Supplementary Data Fig. 6) compared with the particulate SVOC fraction based on the MABNAG model results (Fig. 4a).

## Data availability

The data sets generated during and/or analyzed during this study are available from the corresponding authors on reasonable request. The data relevant to reproducing the figures in the paper are publicly available on the database of the Bolin Centre for Climate Research (https://bolin.su.se/data/).

## Code availability

Codes are available upon reasonable request from the corresponding authors.

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

## Acknowledgements

The authors thank the staff at Hyytiälä for their help during the campaign. Academy of Finland Center of Excellence programme (grant no. 307331) and Academy of Finland grant no. 299544 (T.Y., A.H.), grant no. 304347 (T.P.). European Commission (ACTRIS2, iCUPE (M.K., T.P.)). European Research Council (ATM-GTP (M.K.)). US National Science Foundation grant AGS1801897 (N.M.D.). Department of Energy (DE-SC0006867 and DE-SC0018221 (J.A.T.), BAECC (T.P.)).

## Author contributions

J.A.T., C.M., T.P., and F.D.L. designed the field deployment. F.D.L., C.M., A.L., and J.H. performed the measurements. C.M., F.D.L., A.L., and J.A.T. performed data analysis. T.Y. and A.H. performed the modeling. C.M., J.A.T., T.Y., I.R., A.H., N.M.D., M.K., and M.H. were involved in the scientific interpretation and discussion. C.M., J.A.T., T.Y., A.H., and I.R. wrote the paper. All authors commented on the paper.

## Competing interests

The authors declare no competing interests.
