## [Peer Review File · Nature Communications]

Reviewers' comments:

Reviewer #1 (Remarks to the Author):

The revised paper is improved over the previous version I reviewed. This is a paper describing novel measurements and a model used to interpret them that would be of significant interest to my community. The measurements and prior work on which the modeling is based is of high quality and while limited in some ways (as noted in the manuscript and below) it is not flawed. The strong points of the manuscript remain, and some of the problems have been addressed in part -- but some issues from my initial comments remain. Below are my additional comments, which to my view still require clarification of meaning, terminology, and results before publication is considered. I think there may have been a bit of a rush in this revision, as some of the writing does not seem quite polished.

1. Interpretation of Growth: Do you mean Kelvin effect or Kelvin factor?

I thank the Authors for removing the undefined term "Kelvin factor". I agree SD5 shows the effect of surface tension on vapor pressure over a curved surface, if that is what the "=" in the SD5 caption is supposed to mean. I'm not sure the plot of this standard quantity adds value without giving the explicit model equations in which it appears. Is the model published and available for peer review?

2. Statistical Analysis of Results:

I thank the Authors for adding the ratios of modeled/measured GR (averaged over what period for each case?). They appear to range from 0.7 to 5.9, but they are described as within a factor of 2. How is a factor of almost 6 within a factor of 2? Are these the same numbers shown in the left column of SD3? If no, how are they different? I don't see numbers in SD3 at more than 4:1...so I'm confused what the difference is.

3. Uncertainty Calculation of 53%: This calculation seems based on arbitrarily chosen contributions rather

65 than calculated standard errors.

66 The different terms in the overall uncertainty calculation (following an approach in a peer-reviewed

67 publication) were based on data and background measurements during the campaign, and therefore not

68 arbitrary at all. We have emphasized this in the respective section of the text.

I thank the Authors for “emphasizing” this more in the text -- Where is this in the text? Is it at line 380? If so, it doesn’t address my comment. I was not asking for emphasis, I just want to know what the numbers are that went into the calculation of 53% and where each comes from. And what are “regular” background conditions? I’m afraid this is not well defined terminology. If it comes from a different year or different location, what is the basis for arguing it applies to this study? Providing that rational basis for using the numbers cited from other studies would be the way to show that the choice was not arbitrary, but simply saying it is not arbitrary does not address the concern.

Minor

1. I’m not sure what a “growth abortion” is; I’m hoping it is a typo! If not, please consider rephrasing this terminology to use a term that has less loaded cultural implications.

2. Closure is usually used to mean consistency among independently measured data sets; this seems to be model-msmt comparison which could be either called model evaluation or model validation, depending on your perspective.

3. Three quarters of Figures 3 and SD1,6 seems blank.

4. Are these data publicly available? Where?

5. “we quantitatively explain” – I think within a factor of two is, at best, SEMI-quantitative.

6. I find the use of the word accretion here to lack a clear meaning.

Reviewer #2 (Remarks to the Author):

The authors presented molecular-level measurements of a distribution of organic vapors in a forest environment (Hyytiälä, Finland) during April 11 – June 3, 2014. They identified molecular compositions of >1000 oxygenated volatile organic compounds (OVOC) containing 1 to 30 carbon atoms, up to 17 oxygen atoms, and 0 to 2 nitrogen atoms. The authors further estimated the saturation concentration (C_{sat}) based on molecular formula and then calculated the growth rate of 3-50 nm nanoparticles with a particle growth model. The authors showed that the condensation of low or extremely low volatile organic compounds (LVOC or ELVOC, respectively) dominate the growth of nanoparticles, and concluded that the observed volatility distribution of organic vapors is sufficient to explain nanoparticle growth without invoking various particle-phase processes, such as acid-base reactions or oligomerization.

This manuscript was previously submitted to Nature-Geo. The authors have revised the manuscript in response to the comments and resubmitted it to Nature Communications.

Overall, this manuscript is clearly written and the molecular compositions of OVOC observed during new particle formation (NPF) events provide useful information with regard to the contributions of OVOC to particle growth. The authors have addressed some of issues raised in the previous review and toned down the claim of the significance. However, I am still not sure if the results of the

present study is significant and robust enough for Nature Communication. My further comments are given below.

1. The authors agreed that that OVOC have been well known to be important in particle growth. The authors argued that “The closure we achieve between measured and modelled growth is therefore a key step forward in quantifying vapors in ambient air contributing to particle growth”. The closure is nice but it is based on model simulations of particle growth with uncertainties in a number of parameters (for example, see Supp Data Fig. 3). Also it is not very clear what exactly “a keystone forward” refers to. Do the authors imply that routine measurements of thousands of OVOC should be carried out in the future? Or future particle growth models should be verified with such a closure?

2. One main point the authors tried to highlight is that condensation of OVOC (dominated by LVOC and ELVOC) is sufficient to explain nanoparticle growth without invoking various particle-phase processes. The authors pointed out that “This does not exclude the possibility of particle phase reactions happening, but it does suggest that those reactions are not a major factor driving the growth.” Because of the uncertainty in the growth model and the limitation of the measurements (one season at one site), it is unclear if the results of this study can be extended to other seasons/locations? If not, what is the importance of the present conclusion?

3. With regard to the contribution of SVOC. As pointed out in my previous review report, while C_{sat} for SVOC are larger, the concentrations of SVOC are much higher than those of LVOC and ELVOC (Fig. 1) and SVOC are known to be able to contribute to particle growth through partitioning. Based on AMS measurements in a spring month in Hyytiälä (see Allan et al., 2006; Tsimpidi et al., 2016), the contribution of SVOC to total SOA is equal to that of LVOC. The authors’ response to this comment is copied below:

“Since the AMS is only able to measure particles larger than ~ 70 nm, the references given by the reviewer are not relevant for our study, which focuses on growth of particles to 50 nm. As we state in the manuscript, SVOC with a $C_{sat} > 1 \mu\text{g m}^{-3}$) never reached a gas-phase concentration high enough to contribute significantly to non-reactive condensation. We agree with the reviewer, however, that for larger sizes, the SVOC contribution become more important, as also mentioned related to Supplementary Data Figure 6.”

The argument is not convincing. While the AMS is only able to measure particles larger than ~ 70 nm, it does indicate that a large amount of SVOC can get into the particles in Hyytiälä. Based on Supp data Fig. 5, the effect for SVOC on particles larger than ~ 20 -30 nm is small (< 1.5) and likely not important. Therefore, if SVOC can get into larger particles, they shall be able to get into particles of

20-50 nm as well. The authors argued that “SVOC with a $C_{sat} > 1 \mu\text{g m}^{-3}$) never reached a gas-phase concentration high enough to contribute significantly to non-reactive condensation”. However, SVOC can get into particle through reactive condensation or equilibrium partition even their concentration is lower than C_{sat} . It has been long established that SVOC can get into particles through partitioning (for example, see textbook on Atmospheric Chemistry and Physics by Seinfeld and Pandis). The Supp Data Fig. 6 shows the contribution of SVOC and the authors’ explanation (lines 418-420) appears to be just a speculation (“likely”). To resolve this is critical as it has implication for the robustness of the growth model and conclusion presented in this manuscript.

4. Figure 3 and Supplementary Figure 2. It appears that the modelled GR based on measured OVOC is on average a factor of two larger the observed values. For four out of 13 NPF event days, the over prediction is by a factor of close 4 or larger (Supplementary Figure 2). As the authors pointed out in lines 88-90, the derived vapor concentrations and thus mass fluxes to particles are conservative (lower-limit). Therefore, the over-prediction could be larger. The over prediction will be even larger if the contribution of SVOC is also considered (see Comment #3 above). The significant over-prediction indicates inconsistency and needs to be resolved. In the response to previous comments, the authors offer some possible reasons for the overestimation (e.g, uncertainties in C_{sat} , OVOC concentrations, mass accommodation coefficient, air mass inhomogeneity, etc.). Because of these uncertainties, I feel that this manuscript is more suitable for other specialized journals.

5. Supplementary Figure 2. It will be helpful to the reader if the authors can provide time series of SVOC, LVOC, ELVOC, sulfuric acid, and ammonia you used in the model simulations.

6. Line 243. “The insights provided herein allow for the development of simplified but robust parameterizations ...”. Please give some details on what do you mean “simplified but robust parameterizations”.

MANUSCRIPT NCOMMS-19-05099-T: POINT-BY-POINT RESPONSES TO REVIEWERS' COMMENTS

Claudia Mohr, Joel A. Thornton, Arto Heitto, Felipe D. Lopez-Hilfiker, Anna Lutz, Ilona Riipinen, Juan Hong, Neil M. Donahue, Mattias Hallquist, Tuukka Petäjä, Markku Kulmala, Taina Yli-Juuti

Reviewers' comments:

Reviewer #1 (Remarks to the Author):

The revised paper is improved over the previous version I reviewed. This is a paper describing novel measurements and a model used to interpret them that would be of significant interest to my community. The measurements and prior work on which the modeling is based is of high quality and while limited in some ways (as noted in the manuscript and below) it is not flawed. The strong points of the manuscript remain, and some of the problems have been addressed in part -- but some issues from my initial comments remain. Below are my additional comments, which to my view still require clarification of meaning, terminology, and results before publication is considered. I think there may have been a bit of a rush in this revision, as some of the writing does not seem quite polished.

RE: We thank the reviewer for the very positive assessment of the manuscript. We have now carefully revised the manuscript based on the suggestions by the reviewers, including paying particular attention to the terminology, wording and style.

1. Interpretation of Growth: Do you mean Kelvin effect or Kelvin factor?

I thank the Authors for removing the undefined term “Kelvin factor”. I agree SD5 shows the effect of surface tension on vapor pressure over a curved surface, if that is what the “=” in the SD5 caption is supposed to mean. I’m not sure the plot of this standard quantity adds value without giving the explicit model equations in which it appears. Is the model published and available for peer review?

RE: Indeed SD5 shows the effect of surface tension (and hence particle size) on the equilibrium vapor pressure, as calculated within MABNAG (see Ref 26 for the details of the calculation). We have, however, removed this plot from the revised manuscript as it is not necessary for conveying the key messages of the study.

2. Statistical Analysis of Results:

I thank the Authors for adding the ratios of modeled/measured GR (averaged over what period for each case?). They appear to range from 0.7 to 5.9, but they are described as within a factor of 2. How is a factor of almost 6 within a factor of 2? Are these the same numbers shown in the left column of SD3? If no, how are they different? I don’t see numbers in SD3 at more than 4:1...so I’m confused what the difference is.

RE: The ratios of modelled/measured GR in SD2 are calculated from average modelled and measured GR over the entire period of the respective simulation ($3 \text{ nm} < d_p < 30 \text{ nm}$) or measured growth. As written in lines 358 – 361 of the manuscript, start times for the simulations were chosen as the times during an NPF event when the total concentration of 3-5 nm particles measured by the DMPS was highest, and stopped

at 99400 s or earlier in case the particle diameter reached 52 nm. Given the differences in growth rates for the different NPF events, the time periods over which growth was simulated and subsequently averaged can thus vary. As shown in Figure 3, there are significant uncertainties in both measured and modelled GR, and for most of the NPF events the ratio of modelled/measured GR is between 0.5 and 2 within the uncertainties. We added the following information to the figure caption of SD2: “[...] (*averaged over the entire period of the simulation run*).”

The median of all modelled/measured growth rates of all events (corresponding to the numbers in SD2 and Figure 3) is 2.1, which we rounded down to 2 given the lack of precision in this context.

The numbers in SD2 correspond to the values shown in Figure 3. This is the base case of our model runs, using the *Csat* parameterization given in line 330 in the manuscript. In order to clarify this we have modified the last sentence in the caption of SD2 as following: “*The numbers indicate the ratio of modelled to measured growth rates shown in Figure 3 [...]*.”

To demonstrate the sensitivity of our results to the *Csat* parameterization (and eventually its limitations), gas phase concentrations, and to the value of the mass accommodation coefficient, we had added SD3 to the manuscript. Since our base case is generally biased high regarding growth rates (as shown in Figure 3), we used the upper limits of the *Csat* values, the lower limit of the gas phase concentrations, as well as the lower limit of the mass accommodation coefficient for the runs displayed in SD3. This is the reason why the numbers in SD3 do not exceed ratios of 4:1. We have no indication of a systematic bias of the measured OVOC beyond our stated calibration uncertainty, or proof of an accommodation coefficient lower than unity (as we state in manuscript lines 142 – 144), therefore, as mentioned in lines 138 – 139, uncertainties stemming from *Csat* parametrizations significantly affect the mean bias between modelled and measured growth rates (as shown by the model results in SD3a, where we increased *Csat* arbitrarily by a factor of 100, and in SD3b, where we used a modified version of our parametrization that was tuned to laboratory data (Stolzenburg et al., 2018)). What makes our study unique in this regard and the results therefore more robust is the fact that the *Csat* parametrization used here was not tuned to our dataset, and is therefore completely independent of the compounds measured.

3. Uncertainty Calculation of 53%:

This calculation seems based on arbitrarily chosen contributions rather than calculated standard errors. The different terms in the overall uncertainty calculation (following an approach in a peer-reviewed publication) were based on data and background measurements during the campaign, and therefore not arbitrary at all. We have emphasized this in the respective section of the text.

I thank the Authors for “emphasizing” this more in the text -- Where is this in the text? Is it at line 380? If so, it doesn’t address my comment. I was not asking for emphasis, I just want to know what the numbers are that went into the calculation of 53% and where each comes from. And what are “regular” background conditions? I’m afraid this is not well defined terminology. If it comes from a different year or different location, what is the basis for arguing it applies to this study? Providing that rational basis for using the numbers cited from other studies would be the way to show that the choice was not arbitrary, but simply saying it is not arbitrary does not address the concern.

RE: In our response to the reviewer’s initial comments we referred to lines 372 – 376 for the description of the OVOC concentration uncertainties. In the following we will provide more details regarding the calculation of uncertainties that will hopefully fully address the reviewer’s concerns. We do not mention “regular background conditions”, but “regular background measurements”. By this we mean that during

the measurement campaign we overflowed the critical orifice of the gas-phase inlet of the I-TOF-CIMS with UHP N₂ for 10 seconds every 5 minutes during the gas-phase measurement period, as stated in the manuscript in lines 298 - 300. So the background measurements were done in-situ, with the same instrument, and during acquisition of the data used in this study.

In the publication by Thomson et al., 2017, to which we refer to, the reader can find a detailed description of the individual terms (precision sensitivity, accuracy sensitivity, precision background, accuracy background, precision signal, accuracy signal) that go into the calculation of the total uncertainties for an I-TOF-CIMS. We therefore do not regard it as necessary to add such details here, however emphasize that the data used to calculate the uncertainties in our study come from the measurements and background measurements from this instrument and this campaign.

We have modified the corresponding section as following: *“Uncertainties of OVOC mass concentrations were calculated following the procedure detailed in Thompson et al., 2017. In general, gas phase concentrations of the I-TOF-CIMS are determined as the difference between the signal and background multiplied by sensitivity. Each of the three components has an accuracy and a precision contributing to the overall uncertainty. For the data presented here, signal accuracy (determined from regular calibrations using formic acid) and signal precision (standard deviation of steady signal) were 17 and 9%, respectively; background accuracy and precision (determined using the background measurements regularly performed during the campaign) 40 and 12%, respectively; sensitivity accuracy and precision (determined from repeated calibrations of formic acid) 18 and 20%, respectively, resulting in an overall uncertainty of 53%.”*

Minor

1. I'm not sure what a “growth abortion” is; I'm hoping it is a typo! If not, please consider rephrasing this terminology to use a term that has less loaded cultural implications.

RE: We agree with the reviewer and have replaced “growth abortion” by “growth termination”.

2. Closure is usually used to mean consistency among independently measured data sets; this seems to be model-msmt comparison which could be either called model evaluation or model validation, depending on your perspective.

RE: The way we see it, our study presents a *mass* closure. The growth of new particles from a few nanometers to 30 nanometers represents an increase in particulate mass from an *unexplained* set of vapors. This gap in explained mass is what we try to “close.” We use measured aerosol particle size distributions to determine that increase in mass during a growth event, and measured gas and particle phase compositions and volatilities to provide the explanation of vapor mass that explains the particle mass growth. The independent modeling framework, based on the state-of-the-art theoretical understanding of the problem at hand, is needed to robustly account for vapors, which theory predicts condense across the size distribution and growth event, and thus contribute to closure, as it is not possible to measure directly individual vapors colliding with individual nanoparticles. We therefore would prefer to keep the wording as it is.

3. Three quarters of Figures 3 and SD1,6 seems blank.

RE: We will double-check the pdf to make sure the conversion from word works.

4. Are these data publicly available? Where?

RE: We are not sure to what data the reviewer refers to here, but in general all the data are available upon request from the corresponding authors. *Nature Communications* also requires now that data of a published manuscript's figures are provided in a separate source file, in an effort to ensure reproducibility of research data.

5. “we quantitatively explain” – I think within a factor of two is, at best, SEMI-quantitative.

RE: We understand the reviewer's desire to have a *better* quantitative closure, but our explanation of the growth is quantitative. We present quantitative data of both measured and modelled growth rates, and evaluate their agreement quantitatively, thus we do consider the study to be quantitative – particularly given the uncertainties inherent with any field measurements. There have been many studies that claim “X or Y drive the growth” but do not quantitatively evaluate how well vapor measurements support that claim.

6. I find the use of the word accretion here to lack a clear meaning.

RE: We use the term “accretion” as defined in Barsanti et al., 2004, who introduced it to describe combination reactions of atmospheric organic molecules increasing their mass. We have added this reference to the manuscript.

Reviewer #2 (Remarks to the Author):

The authors presented molecular-level measurements of a distribution of organic vapors in a forest environment (Hytylää, Finland) during April 11 – June 3, 2014. They identified molecular compositions of >1000 oxygenated volatile organic compounds (OVOC) containing 1 to 30 carbon atoms, up to 17 oxygen atoms, and 0 to 2 nitrogen atoms. The authors further estimated the saturation concentration (C_{sat}) based on molecular formula and then calculated the growth rate of 3-50 nm nanoparticles with a particle growth model. The authors showed that the condensation of low or extremely low volatile organic compounds (LVOC or ELVOC, respectively) dominate the growth of nanoparticles, and concluded that the observed volatility distribution of organic vapors is sufficient to explain nanoparticle growth without invoking various particle-phase processes, such as acid-base reactions or oligomerization.

This manuscript was previously submitted to Nature-Geo. The authors have revised the manuscript in response to the comments and resubmitted it to Nature Communications.

Overall, this manuscript is clearly written and the molecular compositions of OVOC observed during new particle formation (NPF) events provide useful information with regard to the contributions of OVOC to particle growth. The authors have addressed some of issues raised in the previous review and toned down the claim of the significance. However, I am still not sure if the results of the present study is significant and robust enough for Nature Communication. My further comments are given below.

RE: We thank the reviewer for the positive comments and hope for our data to speak for themselves and prove that they merit publication in Nature Communications. We emphasize again that this is the first study from the field presenting closure of measured and modelled particle growth rates (based on measured concentrations of the OVOC compounds used as model input) without tuning the model to the data, in contrast to earlier studies (based on laboratory data only) published in high impact journals (e.g. Tröstl et al., 2016, Stolzenburg et al., 2016).

1. The authors agreed that that OVOC have been well known to be important in particle growth. The authors argued that “The closure we achieve between measured and modelled growth is therefore a key step forward in quantifying vapors in ambient air contributing to particle growth”. The closure is nice but it is based on model simulations of particle growth with uncertainties in a number of parameters (for example, see Supp Data Fig. 3). Also it is not very clear what exactly “a key step forward” refers to. Do the authors imply that routine measurements of thousands of OVOC should be carried out in the future? Or future particle growth models should be verified with such a closure?

RE: Indeed the focus here is on the quantification of the vapors in ambient air contributing to particle growth. Somewhat repeating what we have said in the response to the previous comment, we list here the reasons why this study represents a key step forward:

1. It presents the first closure of measured and modelled particle growth using field data, which are much more complex than the proxy systems used in the laboratory;
2. It is based on quantitative measurements of OVOC species used as input into the growth model, without the model parameters being tuned to fit the measured growth rates. It thus represents the first identification of the molecular formulae of compounds explaining particle growth in ambient air. Before this was only achieved for laboratory data, and with invoking an important role of non-identified and not measured LVOC (Tröstl et al., 2016), or tuning *C_{sat}* to reproduce measured growth rates (Stolzenburg et al., 2018));
3. We also emphasize here that the resulting particle-phase composition (dry mass increase as a function of particle size, Figure 4a) does not change significantly with the different model runs (SD3). Obviously, in environments less dominated by biogenic emissions, or environments where isoprene emissions dominate over monoterpene emissions, the molecular formulae of the compounds contributing to particle growth can be different. However, compounds need to be in a similar range of volatility (ELVOC – LVOC), which can help constrain particle growth also in other environments and other seasons, as we write in our conclusions (lines 233 – 243).

The reviewer makes a good point about the routine measurements of thousands of OVOC that should be carried out in the future – we indeed believe that this will become reality within a decade, as these instruments become more widely used, as is already evident now from the increasing number of field campaign datasets (e.g. Huang et al., 2018, Huang et al., 2019). Furthermore, our study validates the use of the MABNAG-type models for accurately predicting ultrafine particle growth without any additional processes in environments where estimates of the gas-phase composition are available.

Based on the reviewer’s suggestion we have added the following sentence to the conclusions section: *“With the availability of highly chemically and temporally resolved datasets such as ours likely to increase in the near future, our study will help constrain and understand growth also in other environments than the one described here.”*

2. One main point the authors tried to highlight is that condensation of OVOC (dominated by LVOC and ELVOC) is sufficient to explain nanoparticle growth without invoking various particle-phase processes. The authors pointed out that “This does not exclude the possibility of particle phase reactions happening, but it does suggest that those reactions are not a major factor driving the growth.” Because of the

uncertainty in the growth model and the limitation of the measurements (one season at one site), it is unclear if the results of this study can be extended to other seasons/locations? If not, what is the importance of the present conclusion?

RE: We appreciate the reviewer's concern, but believe that we are careful with our language in conclusions throughout the paper to not imply they necessarily extend beyond our data sets. In regards to the specific point about the role of particle-phase processes, we feel it is important, given difficulties explaining growth in past measurements, to point out here that we are able to account for the observed growth without invoking them, merely as a clear statement of fact of what we did and did not do in the analysis.

That said, to some extent we have elaborated on the transferability of our study to other locations/other seasons ("other environments"), and the robustness of the resulting particle phase composition as a function of particle size, under point 3 of the response to the previous comment. Further datasets with detailed chemical composition measurements of OVOC during particle growth events may become available, however, we argue that there will always be limitations in terms of locations and seasons covered, as is the case for any atmospheric observation. The benefits of the current location and timing is that there exists a wealth of scientific literature on the measurement station of this study, including long-term measurements of various parameters (e.g. Dal Maso et al., 2005, Hellén et al., 2018), which give indications of the representativeness of conditions prevailing during our study. As the goal of the current study was to understand particle growth during NPF, we chose spring time for our measurements as this is when NPF events are observed most often at this location (Dal Maso et al., 2005).

3. With regard to the contribution of SVOC. As pointed out in my previous review report, while C_{sat} for SVOC are larger, the concentrations of SVOC are much higher than those of LVOC and ELVOC (Fig. 1) and SVOC are known to be able to contribute to particle growth through partitioning. Based on AMS measurements in a spring month in Hyytiälä (see Allan et al., 2006; Tsimpidi et al., 2016), the contribution of SVOC to total SOA is equal to that of LVOC. The authors' response to this comment is copied below:

"Since the AMS is only able to measure particles larger than ~70 nm, the references given by the reviewer are not relevant for our study, which focuses on growth of particles to 50 nm. As we state in the manuscript, SVOC with a C_{sat} > 1 µg m⁻³) never reached a gas-phase concentration high enough to contribute significantly to non-reactive condensation. We agree with the reviewer, however, that for larger sizes, the SVOC contribution become more important, as also mentioned related to Supplementary Data Figure 6."

The argument is not convincing. While the AMS is only able to measure particles larger than ~70 nm, it does indicate that a large amount of SVOC can get into the particles in Hyytiälä. Based on Supp data Fig. 5, Kelvin effect for SVOC on particles larger than ~20-30 nm is small (<1.5) and likely not important. Therefore, if SVOC can get into larger particles, they shall be able to get into particles of 20-50 nm as well. The authors argued that "SVOC with a C_{sat} > 1 µg m⁻³) never reached a gas-phase concentration high enough to contribute significantly to non-reactive condensation". However, SVOC can get into particle through reactive condensation or equilibrium partition even their concentration is lower than C_{sat}. It has been long established that SVOC can get into particles through partitioning (for example, see textbook on Atmospheric Chemistry and Physics by Seinfeld and Pandis). The Supp Data Fig. 6 shows the contribution of SVOC and the authors' explanation (lines 418-420) appears to be just a

speculation (“likely”). To resolve this is critical as it has implication for the robustness of the growth model and conclusion presented in this manuscript.

RE: We see the reviewer’s point and were perhaps not as clear. First, we would like to point out that the process model used here (MABNAG) explicitly includes also the process of what the reviewer refers to as “partitioning” – i.e. condensation driven by a non-zero equilibrium vapor pressure of the organic vapors over the growing particles. Our meaning was that without reactive-condensation, the concentrations of SVOC were not high enough to contribute significantly to initial growth during the events, not that they don’t contribute at all to the background accumulation mode mass which is what an AMS would measure.

We would like to emphasize that one needs to be careful about prior claims regarding aerosol volatility. The paper from Allan et al., 2006, contains no information about SVOC from a volatility measurement perspective, and the study by Tsimpidi et al., 2016, compares a VBS model (containing a highly simplified oxidation scheme driving constituents down by two decades in one or two steps of oxidation from IVOC to SVOC to LVOC) to an AMS factor analysis in which the two factors were named “LV-OOA” and “SV-OOA”. The SV-OOA factor has often been observed to correlate relatively well with (demonstrably semi-volatile) nitrate, but the SV-OOA is understood to represent a broad class of species with a wide volatility range (this is “OOA-2” in Huffman et al., 2009, who examined this factor for Mexico City using a Thermodenuder). Cappa et al., 2010 in turn fit the data from Huffman et al., 2009 to a VBS model and obtained a volatility distribution for SV-OOA with most of the observed mass falling in what we would now classify as the LVOC range.

Pierce et al., 2011 looked directly at the opposite question ($1 - f_{\text{SVOC}}$, where f_{SVOC} is the fraction of SVOC contributing to growth in Hyytiälä), and concluded that well over 50% of the condensing organics were not SVOC. In this analysis the most volatile organics were of $C^* = 0.1 \mu\text{g m}^{-3}$, and all simulations with significant fractions of $0.1 \mu\text{g m}^{-3}$ material failed to reproduce observed growth. It is therefore not clear that experiments in the literature conclusively show that a large fraction of the OA in Hyytiälä are what we now call SVOC ($1 < C^* < 1000 \mu\text{g m}^{-3}$).

The reviewer is correct that above even 10 nm the Kelvin effect is secondary to the volatility distribution itself, so SVOC will be equally important in 10-20 nm particles as in 100-700 nm particles, all else being equal. Provided that there are not significant diffusion limitations within the particles and that the mass accommodation coefficient is near 1, the equilibration timescale for SVOC should be equal to the condensation sink (10-20 min here) and so the fractional contribution of SVOCs to the particle mass and the particle growth rates above 10 nm should simply be their aggregate saturation ratio. Ultimately, the balance of the evidence shows that their mass fraction is small (well under 50%) and so their contribution to growth is equally small.

We also plot here for the reviewers’ and editor’s information more data from the 2013 campaign (compare SD5 and Methods). Figure R1a show a comparison of H:C and O:C ratios of AMS and CIMS during the entire spring campaign 2013, during NPF events only, and during measurements with a MOUDI impactor at the top of the CIMS inlet, removing particles larger than ~56 nm. Without the MOUDI, CIMS and AMS elemental ratios overlap; with the MOUDI, the mean O:C and H:C ratios are higher than what the AMS measures, indicating a bigger fraction of larger and therefore less volatile molecules in the particles with sizes 60 nm and smaller (Figure R1b). This supports that there is a particle size effect on the chemical composition, O:C ratio, and ultimately volatility of the compounds, not due to the Kelvin effect, but the difference in origin of the particles of different sizes.

Figure R1: Elemental ratios of bulk organic aerosol measured with AMS and an acetate TOF-CIMS in Hyytiälä during spring 2013 during the entire campaign, during NPF events only, and during periods when the acetate TOF-CIMS inlet had a MOUDI at the top of the particle-phase inlet (a), and resulting classification of the organic aerosol of the respective periods in volatility groups (b).

4. Figure 3 and Supplementary Figure 2. It appears that the modelled GR based on measured OVOC is on average a factor of two larger than the observed values. For four out of 13 NPF event days, the over-prediction is by a factor of close to 4 or larger (Supplementary Figure 2). As the authors pointed out in lines 88-90, the derived vapor concentrations and thus mass fluxes to particles are conservative (lower-limit). Therefore, the over-prediction could be larger. The over-prediction will be even larger if the contribution of SVOC is also considered (see Comment #3 above). The significant over-prediction indicates inconsistency and needs to be resolved. In the response to previous comments, the authors offer some possible reasons for the overestimation (e.g., uncertainties in C_{sat} , OVOC concentrations, mass accommodation coefficient, air mass inhomogeneity, etc.). Because of these uncertainties, I feel that this manuscript is more suitable for other specialized journals.

RE: We politely disagree here, because for the past two decades, there has been more than an order of magnitude discrepancy between the observed particle growth rates at this location (and other forested sites) and the measured vapors potentially driving growth. Demonstrating that we can now provide measurements that explain this growth to within a factor of 2 on average through multiple events on different days and conditions, is a novel advance of our *fundamental* understanding of new particle formation and growth in the real atmosphere (not a simplified or idealized experiment).

That is, in the context of an existing order of magnitude or more deficit in our understanding of new particle mass sources, a factor of 2 error or mean bias is a relatively small problem for the first comprehensive assessment of current measurement capabilities. That said, our paper also further illustrates what our state-of-knowledge is on C_{sat} and how important refining our understanding of its estimation will be going forward. This paper will thus spur new research activities aimed at this objective.

5. Supplementary Figure 2. It will be helpful to the reader if the authors can provide time series of SVOC, LVOC, ELVOC, sulfuric acid, and ammonia you used in the model simulations.

RE: We have added the time series of SVOC, LVOC, ELVOC, sulfuric acid, and ammonia to Supplementary Data Figure 2. As written in the manuscript, lines 354 – 356, for NPF events where concentration measurements of ammonia and sulfuric acid were missing, daytime averages (8 am – 6 pm) over the whole measurement period were used.

6. Line 243. “The insights provided herein allow for the development of simplified but robust parameterizations ...”. Please give some details on what do you mean “simplified but robust parameterizations”.

RE: We have now modified this to be just “robust parameterizations”. The fact that we have a process model with such a predictive power over measurements collected on a field site that can be considered representative of much of the boreal zone, allows for using this model for simplified descriptions of the growth process with e.g. a smaller number of organic tracers.

References cited in the responses:

- Stolzenburg, D., et al., Proc. Natl. Acad. Sci. U.S.A. 2018, 115 (37) 9122-9127.
Thompson, S. L., et al., Aerosol. Sci. Technol. 2017, 51 (1), 30-56.
Barsanti, K. et al., Atmos. Environ. 2004, 38 (26), 4371-4382.
Tröstl, J., et al., Nature 2016, 533 (7604), 527-531.
Huang, W., et al., Environ. Sci. Technol. 2019, 5, 33, 1165-1174.
Huang, W., et al., Atmos. Chem. Phys. Disc., 2019.
Dal Maso, M., et al., Boreal Environ. Res. 2005, 10 (5), 323-336.
Hellén, H., et al., Atmos. Chem. Phys. 2018, 18, 13839-13863.
Allan, J. D., et al., Atmos. Chem. Phys. 2006, 6, 315-327.
Tsimpidi, A. P., et al., Atmos. Chem. Phys. 2016, 16, 8939-8962.
Huffman, J. A., et al., Atmos. Chem. Phys. 2009, 9, 7161-7182.
Cappa, C. D., et al., Atmos. Chem. Phys. 2010, 10, 5409-5424.
Pierce, J. R., et al., Atmos. Chem. Phys. 2011, 11, 9019-9036.

Reviewers' comments:

Reviewer #1 (Remarks to the Author):

As with my first review, I really like the impressive chemical measurements and the comparison of them to other measured properties as well as model simulations. Given that, I really wanted to like this revised version, especially since the clarifications and changes I requested did not, as far as I can tell, in any way detract from the substance of their results but only from its packaging into things that are beyond what is shown to be true. So I was surprised at the amount of seemingly pointless pushback without supporting evidence. While this is clearly a question that the Editor can adjudicate, I infer that it was sent back to me because he would like to know my opinion. I therefore further explain the points raised in my previous reviews below and how the authors' responses are not sufficient to address them.

1. Following up on the responses to my earlier comments, it is true that the Kelvin effect says that for components that are not non-volatile (including volatile and semi-volatile) that the equilibrium vapor pressure is highest for the smallest particles. This tends to mean that such vapors condense preferentially on larger particles. The role of particle-phase reactions is thus to eliminate this effect by making some vapors effectively non-volatile. Thus for components that undergo such reactions to become NV, it is not the KE that makes them important at small sizes but the ABSENCE of a KE. Important clarifications are needed to this explanation to make this sentence make sense.

HERE IS A SUGGESTED WAY TO DO THIS THAT IS NOT PERFECT BUT AT LEAST GETS RID OF THE PROBLEMS WITH ATTRIBUTING REDUCED SIZE DEPENDENCE TO KE:

"The ELVOC are calculated to make larger contributions THAN MORE VOLATILE GASES at the very early stages of particle growth due to THE REDUCTION of the Kelvin effect WITH THE REDUCED/LOW VOLATILITY OF REACTING GASES, ..." [ADD CITATION TO 26].

To follow up on this point, the terminology of Kulmala et al. 2013 cited here (and reference to Kulmala et al. 2004 therein) referring to the role of NV components as nano-Kohler is really a contradiction in terms, to the extent to which it refers to the lack of a kelvin effect for nonvolatile gases, since the premise of kohler is some volatility of the condensing vapor, as is inherently true of KE also. There is no change in equilibrium vapor pressure with size for a component that has a vapor pressure of zero, and a component with $EVP=0$ has no Kelvin effect (and, without a KE, the Kohler curve is just the Raoult effect so nano-Kohler is then just Raoult's effect). I realize that some may consider this a minor matter of semantics (or, perhaps less charitably, buzz word hype), but it is not; the issue is the basic physics to which the mechanism is attributed.

The previously included plot showing EVP as a function of size was misleading, since the high values at small sizes is actually what reduces the driving force for condensation, so I don't see the point in including that. But the description does need to have the principle correct, i.e. that KE means there is less condensation at small sizes not more.

It also strikes me that the only plots that show the size dependence discussed are the model results, i.e. there are no size dependent composition measurements. So I think the attribution of this as a reason should be couched in somewhat more speculative terms, since it provides a sufficient explanation but not necessarily the only one.

2. I still don't see what is within a factor of 2. For the averaged values shown, 4 of 10 shown are between 0.5 and 2. The rest are not. If you mean the range is larger than 0.5 to 2 because of some uncertainties, please say what uncertainties and what the range is. Is the idea that they range between $0.5 \pm 50\%$ to $2 \pm 50\%$ (which seems to me to be 0.25 to 3 not to 6? Am I looking at the wrong numbers or is your definition of a factor of 2 not 0.5 to 2? A large part of the previous response seemed tangential or irrelevant to this question.

3. I thank the authors for these clarifications. They are a big improvement.

MINOR but still should be fixed:

1. The new term of growth "termination" – I thank the authors for removing the term "abortion". However, this is not a correct way to characterize the observation of "short or partial observations of growth events" at a non-Lagrangian observation site. In other words, do you actually know that the growth was terminated or is it just a change in air mass? Field observations cannot be interpreted as a time sequence of the same air mass without careful evidence in support of that assertion, none of which was provided here. I recommend that the authors change their terminology to something like "short or partial observations of growth events" to indicate that they don't know whether or not the event terminated, only that their observation of it ended. Similarly the asserted "requirement for the box model of air masses to be homogeneous" is not quite right; the box model is Lagrangian and the assumption is that the air mass would need to be sufficiently homogeneous on a spatial scale equivalent to the time scale of the growth to make the Eulerian observations pseudo-Lagrangian. So for a 10 hr event the air mass should be homogeneous on scales of say $(36 \times 10^3 \text{ m/s} \times 5 \text{ m/s} \sim 200 \text{ km})$. What evidence is there that this is true in this case? Is there a citation available for prior work showing this? I am not actually saying the authors should not do this, but I am asking that

the authors actually do the relevant work to make the appropriate meteorological case for at least some fraction of the events or at least cite another work that has done this to provide the reader with some basic evidence that the reasoning can be based on (or precedent for it).

2. The reviewers argue in response to my question that they provide a closure rather than a model validation, but they argue to Reviewer 2 that this study has validated the model (for use at other locations). I echo Reviewer 2's concerns about extrapolation beyond the results presented, which I believe was an issue at the first review round. But I also believe this is contradictory to claim it is both a closure and a model validation. The two are different, and the authors are misusing the terms as interchangeable. Again it is important to not over-reach the claims of what has been done. The closure assumes that the model is known to be correct (e.g. for Mie scattering calculations of size distributions), showing that independently measured variables are consistent (but not then "validating" Mie theory, of course). If the model is assumed to be correct, it cannot then be validated. So the authors need to pick one. To me it seems that the stronger science is in showing that the model is sufficient to explain the growth given the measurements, which technically is more showing consistency than validation (for those of us who use terminology consistent with modeling literature). In addition to this inconsistency in their argument, while "closing a gap" is correct English, it does not represent the accepted use of the term closure in the field. The authors are welcome to say they are "closing the gap" in the missing vapors, but calling this a "closure" of measurements (and more so of the authors' extension of the term to be between measurements and model) is misconstruing its meaning in a way that would engender confusion in the field. And it is simply incorrect.

3. Good.

4. I strongly recommend archiving the data and codes and results consistent with FAIR principles, rather than relying on a static archive like a supplement.

5. Thank you for understanding, but really a clarification is what is needed. If you find the term semi-quantitative too vague, then specify that by "quantitative" you mean to within a factor of 2 (which most would consider a non-standard definition so the specification is needed).

6. Thank you for adding this relevant reference but I do hope that the use of the term extends beyond a single publication in order to use it here without definition.

Additional

1. 189 add "of"

2. The other Reviewer has raised a number of issues with which I agree but will not address.

Reviewer #2 (Remarks to the Author):

The authors have addressed my concerns. I recommend the publication of this manuscript in Nature Communications.

MANUSCRIPT NCOMMS-19-05099A: POINT-BY-POINT RESPONSES TO REVIEWERS' COMMENTS

Claudia Mohr, Joel A. Thornton, Arto Heitto, Felipe D. Lopez-Hilfiker, Anna Lutz, Ilona Riipinen, Juan Hong, Neil M. Donahue, Mattias Hallquist, Tuukka Petäjä, Markku Kulmala, Taina Yli-Juuti

Reviewers' comments:

Reviewer #1 (Remarks to the Author):

As with my first review, I really like the impressive chemical measurements and the comparison of them to other measured properties as well as model simulations. Given that, I really wanted to like this revised version, especially since the clarifications and changes I requested did not, as far as I can tell, in any way detract from the substance of their results but only from its packaging into things that are beyond what is shown to be true. So I was surprised at the amount of seemingly pointless pushback without supporting evidence. While this is clearly a question that the Editor can adjudicate, I infer that it was sent back to me because he would like to know my opinion. I therefore further explain the points raised in my previous reviews below and how the authors' responses are not sufficient to address them.

RE: We thank the Reviewer for the positive remarks and hope to have addressed the remaining issues below.

1. Following up on the responses to my earlier comments, it is true that the Kelvin effect says that for components that are not non-volatile (including volatile and semi-volatile) that the equilibrium vapor pressure is highest for the smallest particles. This tends to mean that such vapors condense preferentially on larger particles. The role of particle-phase reactions is thus to eliminate this effect by making some vapors effectively non-volatile. Thus for components that undergo such reactions to become NV, it is not the KE that makes them important at small sizes but the ABSENCE of a KE. Important clarifications are needed to this explanation to make this sentence make sense.

HERE IS A SUGGESTED WAY TO DO THIS THAT IS NOT PERFECT BUT AT LEAST GETS RID OF THE PROBLEMS WITH ATTRIBUTING REDUCED SIZE DEPENDENCE TO KE: "The ELVOC are calculated to make larger contributions THAN MORE VOLATILE GASES at the very early stages of particle growth due to THE REDUCTION of the Kelvin effect WITH THE REDUCED/LOW VOLATILITY OF REACTING GASES, ..." [ADD CITATION TO 26].

To follow up on this point, the terminology of Kulmala et al. 2013 cited here (and reference to Kulmala et al. 2004 therein) referring to the role of NV components as nano-Kohler is really a contradiction in terms, to the extent to which it refers to the lack of a kelvin effect for nonvolatile gases, since the premise of kohler is some volatility of the condensing vapor, as is inherently true of KE also. There is no change in equilibrium vapor pressure with size for a component that has a vapor pressure of zero, and a component with $EVP=0$ has no Kelvin effect (and, without a KE, the Kohler curve is just the Raoult effect so nano-Kohler is then just Raoult's effect). I realize that some may consider this a minor matter of semantics (or, perhaps less charitably, buzz word hype), but it is not; the issue is the basic physics to which the mechanism is attributed.

The previously included plot showing EVP as a function of size was misleading, since the high values at small sizes is actually what reduces the driving force for condensation, so I don't see the point in

including that. But the description does need to have the principle correct, i.e. that KE means there is less condensation at small sizes not more.

It also strikes me that the only plots that show the size dependence discussed are the model results, i.e. there are no size dependent composition measurements. So I think the attribution of this as a reason should be couched in somewhat more speculative terms, since it provides a sufficient explanation but not necessarily the only one.

RE: We fully agree with the Reviewer's definition of the Kelvin effect, and the importance of phrasing our statements in a physically meaningful way. Since reactions in the condensed phase were neglected in our growth calculations (lines 327 – 328), the net increase of a compound's mass in the particle phase is determined by the condensation mass flux driven by the difference between ambient concentration of the vapor and the equilibrium vapor concentration (defined by the saturation concentration, Kelvin term and Raoult's term). What we are simply trying to say in our manuscript is that, due to the Kelvin effect, which makes not non-volatile vapors preferentially condense onto larger particles, we find the largest mass increase fraction of the least volatile compounds (ELVOC) at the smallest sizes (because there is less condensation of the more volatile compounds, as the Reviewer points out above), and as the particles become bigger, the mass increase fractions of the more volatile compounds (LVOC, SVOC) increase. To clarify we have rephrased the corresponding lines 189 – 191 as following: "The SVOC and LVOC are calculated to make smaller contributions at the very early stages of particle growth due to the influence of the Kelvin effect, hence the relative contribution of the ELVOC is largest at the smallest particle sizes. Other work suggests that the ELVOC monomers and dimers play a dominant role in particle nucleation itself³⁶. ELVOC also remain significant contributors to particle mass (30%) up to sizes of 50 nm." We also have taken pains in the revision to make sure that when we state conclusions about size dependence it is based on model results (including adding "Modelled" in line 279, "In our model" in line 193).

2. I still don't see what is within a factor of 2. For the averaged values shown, 4 of 10 shown are between 0.5 and 2. The rest are not. If you mean the range is larger than 0.5 to 2 because of some uncertainties, please say what uncertainties and what the range is. Is the idea that they range between 0.5 +/- 50% to 2 +/- 50% (which seems to me to be 0.25 to 3 not to 6? Am I looking at the wrong numbers or is your definition of a factor of 2 not 0.5 to 2? A large part of the previous response seemed tangential or irrelevant to this question.

RE: We realize we were not exact enough in our description of modelled/measured growth rate. Indeed, the ratios of modelled/measured GR vary between 0.7 and 5.9, with 8 out of 13 events having ratios between 0.7 and 2.1, and 1 event a ratio of 2.5. The median ratio of modelled/measured GR rate for all events is 2.1. We have modified lines 129 – 132 as following: "For the majority of NPF events, the ratio of modelled and measured growth rates lies between 0.7 and 2.1 (median ratio all events: 2.1)."

3. I thank the authors for these clarifications. They are a big improvement.

MINOR but still should be fixed:

1. The new term of growth "termination" – I thank the authors for removing the term "abortion". However, this is not a correct way to characterize the observation of "short or partial observations of growth events" at a non-Lagrangian observation site. In other words, do you actually know that the growth was terminated or is it just a change in air mass? Field observations cannot be interpreted as a time sequence of the same air mass without careful evidence in support of that assertion, none of which

was provided here. I recommend that the authors change their terminology to something like “short or partial observations of growth events” to indicate that they don’t know whether or not the event terminated, only that their observation of it ended. Similarly the asserted “requirement for the box model of air masses to be homogeneous” is not quite right; the box model is Lagrangian and the assumption is that the air mass would need to be sufficiently homogeneous on a spatial scale equivalent to the time scale of the growth to make the Eulerian observations pseudo-Lagrangian. So for a 10 hr event the air mass should be homogeneous on scales of say $(36 \times 10^3 \text{ m/s} \sim 200 \text{ km})$. What evidence is there that this is true in this case? Is there a citation available for prior work showing this? I am not actually saying the authors should not do this, but I am asking that the authors actually do the relevant work to make the appropriate meteorological case for at least some fraction of the events or at least cite another work that has done this to provide the reader with some basic evidence that the reasoning can be based on (or precedent for it).

RE: We have replaced the sentence in lines 143 – 146 by “The event on May 7 exhibits a rather unclear start time, and the event on May 14 represents a short or only partial observation of a growth event.” We also have removed the sentence about the homogeneity of air masses. We agree with the reviewer that the information available is not sufficient to make such a statement (earlier work however has shown that the typical spatial scale of NPF events in this region is 200 – 300 km (Hussein et al., 2009)). Nieminen et al., 2015, showed that the likelihood for an NPF event in Hyytiälä is a complex function of meteorological conditions, air mass, and concentrations of trace gases and particles. Ancillary data from our campaign indicate that May 14 was a day with a likelihood for non-continuous growth. We have added this reference to the manuscript.

2. The reviewers argue in response to my question that they provide a closure rather than a model validation, but they argue to Reviewer 2 that this study has validated the model (for use at other locations). I echo Reviewer 2’s concerns about extrapolation beyond the results presented, which I believe was an issue at the first review round. But I also believe this is contradictory to claim it is both a closure and a model validation. The two are different, and the authors are misusing the terms as interchangeable. Again it is important to not over-reach the claims of what has been done. The closure assumes that the model is known to be correct (e.g. for Mie scattering calculations of size distributions), showing that independently measured variables are consistent (but not then “validating” Mie theory, of course). If the model is assumed to be correct, it cannot then be validated. So the authors need to pick one. To me it seems that the stronger science is in showing that the model is sufficient to explain the growth given the measurements, which technically is more showing consistency than validation (for those of us who use terminology consistent with modeling literature). In addition to this inconsistency in their argument, while “closing a gap” is correct English, it does not represent the accepted use of the term closure in the field. The authors are welcome to say they are “closing the gap” in the missing vapors, but calling this a “closure” of measurements (and more so of the authors’ extension of the term to be between measurements and model) is misconstruing its meaning in a way that would engender confusion in the field. And it is simply incorrect.

RE: The use of the word “validation” in response to Reviewer 2’s concerns was related to the *application* of basic condensation theory (which itself is well established for liquid mixtures of known properties) on atmospheric systems (we wrote the following: “Furthermore, our study validates *the use* of the MABNAG-type models [...]). The use of the word “closure” for the agreement between a measured and modelled value, or between top-down and bottom-up observed quantities is, from our point of view, widely used in the field, e.g. see publications related to the speciated reactive nitrogen budget. However, as we do not think this semantic issue is critical to the communication of our results, we have replaced

closure throughout most of the manuscript. We also fully agree with the reviewer that our results show *consistency* between our model and measurements. Lines 34 – 36 now read as “The agreement between the observed nanoparticle mass growth, and the growth predicted from the observed mass of available condensing vapors in a forested environment thus represents an important step forward in the characterization of atmospheric particle growth”; lines 63 – 66 as “Our agreement in measured nanoparticle mass growth, and the mass of available condensing vapors [...]”; in lines 115 and 142 we have replaced “Closure” by “Agreement”; in line 221 we have removed the word “closure”.

3. Good.

4. I strongly recommend archiving the data and codes and results consistent with FAIR principles, rather than relying on a static archive like a supplement.

We thank the Reviewer for this suggestion, and we will defer to the Editor’s judgement and preference. Much of the metadata is already available online, and we will place the data and codes relevant to reproducing the figures and conclusions in the paper on the database of the Bolin Centre for Climate Research (<https://bolin.su.se/data/>) where they will be publicly available.

5. Thank you for understanding, but really a clarification is what is needed. If you find the term semi-quantitative too vague, then specify that by “quantitative” you mean to within a factor of 2 (which most would consider a non-standard definition so the specification is needed).

RE: Also here we feel like there is a semantic disagreement. We would like to emphasize again that we “present quantitative data of both measured and modelled growth rates, and evaluate their agreement quantitatively, thus we do consider the study to be quantitative”, as stated in the previous response. We have now, however, revised the manuscript as following: In lines 28 – 30, we have removed “quantitatively” from “can quantitatively explain”, but added “quantitative” in front of “molecular level observations”. In lines 58 – 59, we have removed “quantitatively” from “quantitatively explain”, and moved it to before “show”. In this way we hope to emphasize the first point made in this response, that we present quantitative data. Compare also response to major comment 2.

6. Thank you for adding this relevant reference but I do hope that the use of the term extends beyond a single publication in order to use it here without definition.

RE: The term is widely used also in more recent publications (e.g. Zhao et al., 2019; Berndt et al., 2019) and references therein.

Additional

1. 189 add “of”

RE: We have added “of” to “majority of particle growth”.

2. The other Reviewer has raised a number of issues with which I agree but will not address.

RE: The concerns raised by Reviewer 2 have been addressed (see below).

Reviewer #2:

The authors have addressed my concerns. I recommend the publication of this manuscript in Nature Communications.

References cited in the responses:

Zhao, Y., et al., Proc. Natl. Acad. Sci. U.S.A. 2018, 115 (48) 12142-12147.

Berndt, T., et al., Angew. Chem. Int. Ed. 2018, 57, 3820.

Nieminen, T., et al., Atmos. Chem. Phys. 2015, 15 (21), 12385-12396.

Hussein, T., et al., Atmos. Chem. Phys. 2009, 9, 4699-4716.

REVIEWERS' COMMENTS:

Reviewer #1 (Remarks to the Author):

The revised manuscript is acceptable. I recommend it be published.